# CURRICULUM LEARNING AS A TOOL TO UNCOVER LEARNING PRINCIPLES IN THE BRAIN

**Daniel R Kepple**
Friedman Brain Institute
Icahn School of Medicine at Mount Sinai
New York, NY 10029, USA

**Rainer Engelken**
Department of Neuroscience, Zuckerman Institute
Columbia University
New York, NY, 10027, USA

**Kanaka Rajan**
Friedman Brain Institute
Icahn School of Medicine at Mount Sinai
New York, NY 10029, USA

## ABSTRACT

We present a novel approach to use curricula to identify principles by which a system learns. Previous work in curriculum learning has focused on how curricula can be designed to improve learning of a model on particular tasks. We consider the inverse problem: what can a curriculum tell us about how a learning system acquired a task? Using recurrent neural networks (RNNs) and models of common experimental neuroscience tasks, we demonstrate that curricula can be used to differentiate learning principles using target-based and a representation-based loss functions as use cases. In particular, we compare the performance of RNNs using a target-based learning principle versus those using a representational learning principle on three different curricula in the context of two tasks. We show that the learned state-space trajectories of RNNs trained by these two learning principles under all curricula tested are indistinguishable. However, by comparing learning times during different curricula, we can disambiguate the learning principles and challenge traditional approaches of interrogating learning systems. Although all animals in neuroscience lab settings are trained by curriculum-based procedures called shaping, almost no behavioral or neural data are collected or published on the relative successes or training times under different curricula. Our results motivate the systematic collection and curation of data during shaping by demonstrating curriculum learning in RNNs as a tool to probe and differentiate learning principles used by biological systems, over statistical analyses of learned state spaces.

## 1 INTRODUCTION

The biological brain is thought to be the ultimate learner as it learns from few examples, solves unstructured problems, and has an impressive task repertoire. Understanding how it achieves these learning feats could lead us to better artificial intelligence (AI) algorithms (Hassabis et al., 2017; Macpherson et al., 2021). Interrogating learning in the brain, however, poses significant challenges both experimentally and computationally. Experimentally, measuring the entire synaptic connectivity map or connectome–a primary substrate of learning–is only just approaching reality for small brains (Zador et al., 2012; Dorkenwald et al., 2020). Even so, the emerging connectome data provides just a snapshot in time whereas learning concerns connectome dynamics, and these temporal data are still far off. Computationally, it is unclear how a sequence of brain-wide connectivities would be registered in order to reveal how animals learn different tasks (Babai, 2015). Uncovering the key substrates linking structural (connectome), dynamic (neural activity), and behavioral elements in biological brains could help us develop better AI algorithms with brain-like learning properties.

In this paper, we use behavioral dynamics to infer learning principles. Measuring behavior in animals is comparatively easier than neural recordings or synaptic strengths. The challenge, however, is for

theory to attribute individual behaviors to specific learning principles. We suggest the use of curricula to get information from behavioral dynamics about underlying learning principles.

A curriculum is a schedule for information to be presented to a student learner. In the machine learning (ML) framework we consider, the student learner is a neural network, specifically an RNN. Recurrence in network models brings some key advantages we exploit here–ability to produce dynamics and analogy to the biological brain's ubiquitous feedback connections (Yang et al., 2019; Singer, 2021; Ehrlich et al., 2021). Although some benefits of curricula have been shown in ML (Graves et al., 2017; Weinshall et al., 2018; Saglietti et al., 2021), curriculum learning has not been widely adopted for practical applications, with notable exceptions in robotics (James et al., 2019) and reinforcement learning (Taylor & Stone, 2009). In contrast, curriculum learning is very important to experimental neuroscience–animals in neuroscience lab settings are trained using curricula, a process called *shaping* (Pinto et al., 2018; Koay et al., 2021; Guo et al., 2014). Yet, very little relevant neuroscience data on curricula exists, as data are most often collected from "expert" animals after shaping.

Our goal is to classify learners–here, RNNs–based on the principles they use to learn different tasks using a set of pre-designed curricula. We build and analyze RNNs trained on two common experimental neuroscience tasks using three different curricula inspired by shaping procedures. In particular, we use the evidence accumulation task (Pinto et al., 2018; Stine et al., 2020) and delayed decision task (Romo et al., 1999; Constantinidis et al., 2018; Liu et al., 2014). We show, using only behavioral dynamics during the execution of different curricula, that it is possible to distinguish two learning principles solely on the basis of outcomes: target-based learning from learning through representational constraints. Importantly, we find that all RNNs, regardless of curriculum or learning rule, are indistinguishable post-shaping by standard statistical analyses applied to neural dynamics in their trained state. Our results emphasize the importance of studying animals during shaping and the value of curriculum learning in RNNs as a hypothesis test-bed for probing learning principles in the biological brain.

## 1.1 RELATED WORK

Curriculum learning has long been relevant to the fields of AI/ML and neuroscience (Wang et al., 2021). Curricula have been used to learn difficult control problems in robotics and reinforcement learning (Selfridge et al. (1985), Schmidhuber (1991), Sanger (1994)). Elman (1993) noted that humans and animals use curricula and asked how they could benefit machines; and Bengio et al. (2009) has related curriculum learning to input complexity in a key paper on optimizing learning. Despite the presence of curricula in ML literature for several decades, our work is the first we are aware of that uses curricula to characterize learning principles used by neural network models to learn tasks, analogously to experimental animals in lab settings.

Recently, the question of identifying learning rules was studied by Nayebi et al. (2020), where the full knowledge of network activations available to artificial systems is used and related to neural data from experimental recordings. Here, we are interested in being able to glean learning principles solely from behavioral data. Ultimately, our two approaches may be used in tandem to further our understanding of learning principles.

Ashwood et al. (2020) link behavioral data and learning rules using large-scale data from a mouse task in the International Brain Lab (IBL). The IBL task and consequently, the Ashwood et al. (2020) model includes no time dependence, history, or state dependence. Furthermore, mice can learn the task easily without a curriculum. Our work is distinct in that we target complex tasks with temporal dependencies for which a) curricula are appropriate, and b) the analogous experiments require shaping procedures for the lab animal to learn.

In this paper, we demonstrate that different learning principles and curricula converge to similar solutions in all cases (Fig 5). In Maheswaranathan et al. (2019), the authors claim that there is a universal solution for a given task that all learning algorithms converge to. Their approach does not incorporate curricula; our work posits a different, complementary invariance. Furthermore, our goal is not to prove universality, but rather to identify learning principles that may be inaccessible by only studying the trained or post-shaping state.

## 2 MODELS AND METHODS

### 2.1 RECURRENT NEURAL NETWORKS (RNNS)

We use neural networks with $N = 350$ recurrently connected, continuous, firing rate based, leaky integrating model neurons. A neuron with index $n$ connects to a neuron with index $m$ through the recurrent weight $w_{nm}^{\text{rec}}$ and input channel index $j$ through input weight $w_{nj}^{\text{inp}}$. The internal state of neuron $n$, $\mathbf{x}_n$, is determined by:

$$\tau \frac{d\mathbf{x}_n}{dt} = w_{nm}^{\text{rec}} \mathbf{a}_m(t) + w_{nj}^{\text{inp}} \mathbf{v}_j(t) - \mathbf{x}_n \tag{1}$$

where $\mathbf{v}(t)$ is a time-dependent, task-relevant input at time $t$ and $\mathbf{a}(t)$ is the activation function applied to the internal state–here, the hyperbolic tangent function: $\mathbf{a_n}(t) = \tanh(\mathbf{x}_n(t))$. $\tau$ is the time constant of the neuron, here, 10ms. We use the Euler method to calculate neural states with $dt = 1$ms. We define the linear readout of a network at time $t$, $\mathbf{z}(t)$, as the weighted sum of the activations $\mathbf{a}(t)$ via weights $w_n^{\text{out}} : \mathbf{z}(t) = w_n^{\text{out}} \mathbf{a}_n(t)$

Recurrent weights $W^{\text{rec}}$ are initialized i.i.d from a Gaussian with mean 0 and variance $\frac{g^2}{N}$. We set, $g = 1$, the critical point above which random networks are chaotic (Sompolinsky et al., 1988) and backpropagation fails. Input- $W^{\text{inp}}$ and readout weights $W^{\text{out}}$ are each drawn from a uniform distribution from -1 to 1. Internal states, $X$, are initialized from a Gaussian (mean 0, variance 1).

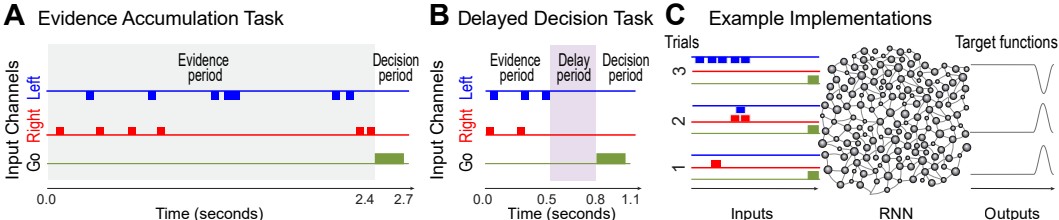

Figure 1: **Task Design.** Both tasks use three input channels. Two correspond to "left" (blue line) or "right" (red line), respectively. The third (green line) sends the go cue, signaling the start of a decision period. **A:** Evidence accumulation (abbreviated EA) tasks contain a 2.4s evidence period (gray box) and 0.3s decision period. During the evidence period, input events or "cues" occur at times generated by a Poisson process in the left/right channels. In the decision period, a target bump reflects the channel with more events during the evidence period. **B:** Delayed decision (abbreviated DD) tasks contains a 0.5s delay period (mauve box) and a 0.5s evidence period. No events are presented during this period. **C:** Example implementations shown here. RNN outputs are compared to task target functions and weights change according to the RNN's learning update.

### 2.2 TASKS

Our tasks model commonly used neuroscience tasks for studying working memory and decision making in rodents, non-human primates, and humans (Pinto et al., 2018; Stine et al., 2020; Romo et al., 1999; Constantinidis et al., 2018). In the rodent setup, a mouse runs down a hall with cues on its left and right. The number of left or right cues inform the mouse which of two turns it should make when the corridor ends, and is rewarded if it turns toward the side with more cues.

In our model tasks, we have a evidence period to mimic the hall in the mouse task and task parameters chosen to be proportional to mouse experiments in Pinto et al. (2018). Two input channels, corresponding to left and right cues in the mouse task, send pulsatile input events into the RNN units, each with an amplitude of 0.25 and duration of 50ms. Cue start times for each channel are Poisson distributed with rates $\lambda$ of $150\text{ms}^{-1}$ and $300\text{ms}^{-1}$ for the two channels throughout. On each simulated trial, $\lambda$s for input channels are randomly swapped. The RNN also has a third input channel which produces a 0.25 amplitude "go cue" lasting $50\text{ms}$ in the decision period. The target function, $z^{\text{tar}}$, for the linearly read out RNN-unit activations is 0 during the cue period. During the decision period, the target function is half a cosine wave starting an ending at zero with amplitude either 2 or -2 depending on which channel accumulated more cues (Fig 1C).

We use two task variants: The first is an evidence accumulation (EA) task (Fig 1A) to challenge primacy and recency biases. RNNs on this task, similar to lab animals, sometimes overweigh early or recent evidence at the cost of running counts and relative discrepancy of cues between the two channels. The second is a delayed decision (DD) task (Fig 1B), with a delay period between the evidence and decision periods to challenge temporal credit assignment, which is often problematic for networks to span.

**Evidence Accumulation (EA) tasks** have a 2.4s cue period followed by a 0.25s decision period.

**Delayed Decision (DD) tasks** have a 0.5s cue period, a 0.5s delay period, and a 0.25s decision period. The target function during the delay is 0 with no cues during this time.

## 2.3 DISCREPANCY

A key characterization of the two task variants is a quantity we call *Discrepancy*. We define discrepancy as the instantaneous difference between the number of cues in the left and the right input channels. Formally, the set of all left event or "cue" times is $\{t_i^{\text{left}}\}_{i=1}^{K^{\text{left}}}$ where $K^{\text{left}}$ is the total number of left events in the trial and $t_i^{\text{left}}$ is the time of the $i$th left event. Discrepancy at time $t$ is defined: $D(t) = \sum_{i=1}^{K^{\text{left}}} H(t - t_i^{\text{left}}) - \sum_{i=1}^{K^{\text{right}}} H(t - t_i^{\text{right}})$ where the $H$ is the Heaviside step function: $H(x) = 1$ if $x > 0$ and 0 otherwise. $D(t) < 0$ when there are more left cues than right. We define *absolute discrepancy* as $|D(t)|$ and the absolute discrepancy of a trial, $|D(t_{\text{final}})|$. For all trials, we enforce $|D(t_{\text{final}})| > 0$.

## 2.4 LEARNING PRINCIPLES

The goal of this paper is to use curriculum learning in RNN models of decision-making tasks as an analogy to shaping in animal neuroscience experiments, in order to identify learning principles employed by the brain. For clarity, we use the following working definitions of learning *principles*, *rules*, and *updates*. We refer to a *learning rule* as the function optimized by learning. *Learning update* defines the trajectory of the learner in this optimization. A *learning principle* is a higher order categorization of learning rules. Examples of learning principles include target learning, maximum entropy, minimum energy, and representational learning, each of which could be implemented by various loss functions (i.e., learning rules) and updates.

We focus on two learning principles of interest to neuroscience to determine whether the brain learns mostly by rewarding and punishing behavioral outputs–target learning, or by enforcing internal representations on its neural dynamics–representational learning (Saxe et al., 2021; Bhand et al., 2011; Yamins et al., 2014). Here, in RNNs trained by learning updates using backpropagation through time, we design separate learning rules to implement *target learning* and *representational learning*.

For **target-based backpropagation (BPT)**, we define an L2 error between a target function for an idealized "behavior-like" output of the task, $\mathbf{z}^{\text{tar}}(t)$ and the RNN's overall output, $\mathbf{z}(t)$:

$$\mathcal{L}^{\text{tar}}(t_{\text{T}}) = \sum_{s=0}^{T} \left[ \mathbf{z}(t_s) - \mathbf{z}^{\text{tar}}(t_s) \right]^2, \tag{2}$$

where $t_T$ is the weight-update time and $t_0$ is the trial start time. $t_T = t_{\text{final}}$, i.e., weights are updated only at the ends of trials.

For **representational backpropagation (BPR)**, we define an L2 error with an idealized representation. For our tasks, the ideal representation can read out which channel has had more evidence. We therefore incorporate discrepancy into the loss function, forcing the network to directly represent the instantaneous difference between the cues in the two input channels throughout a trial. While we only show results for this representational learning rule for simplicity, results in this paper are also consistent for a representational triplet loss or a loss penalizing deviation from two fixed points.

We add representational weights $W^{\text{rep}}$ with elements $w_n^{\text{rep}}$ that linearly read out discrepancy from the RNN units $\mathbf{a}_n$, $\hat{D}(t) = w_n^{\text{rep}} \mathbf{a}_n$. Our representational loss function is then:

$$\mathcal{L}^{\text{rep}}(t_{\text{T}}) = \sum_{s=0}^{T} \left[ \mathbf{z}(t_s) - \mathbf{z}^{\text{tar}}(t_s) \right]^2 + \lambda(t) \left[ \hat{D}(t_s) - D(t_s) \right]^2, \tag{3}$$

where $\lambda(t)$ weights the representational loss. $\lambda(t)$ is 0 during delay and decision periods; representational constraints are only applied during the cue period, during which $\lambda(t_{\text{cue}}) = 0.01$. This parameter was set to bring the discrepancy error to the same order of magnitude to the target error. We note a necessary limitation imposed by using a behavioral matching task is that both models must include target learning principles. Therefore we cannot test the case where the model follows a purely representational learning principle.

Gradients are calculated and accumulated at every time step in parallel trials presented in batch sizes of 32. Weights are updated after each trial using Adam with $\beta_0 = 0.9$ and $\beta_1 = 0.999$ (Kingma & Ba, 2014). Learning rates are chosen proportionally to average weight initialization; 0.01 for readout or output weights, $W^{\text{out}}$ and representational weights $W^{\text{rep}}$, and 0.0003 for recurrent weights, $W^{\text{rec}}$.

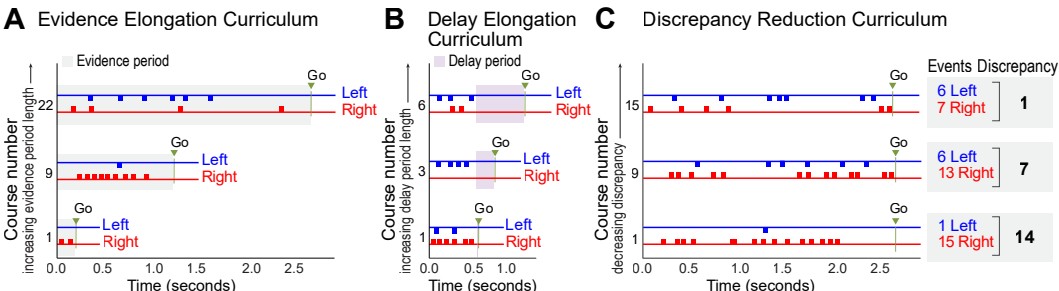

Figure 2: **Curriculum Design.** Each curriculum is presented in "courses", blocks of trials with the same task parameters. The number of trials in a given course is determined by the performance of the network. **A** *Evidence elongation curriculum.* Cue period is elongated at 0.1s per course for both DD and EA tasks (EA schematized here). **B** *Delay elongation curriculum.* Delay period is elongated by 0.1s per course in DD tasks. **C** *Discrepancy reduction curriculum.* The minimum allowable discrepancy is decreased by 1 for each new course, until all discrepancies are presented.

## 2.5 CURRICULA

We define curricula using a hierarchical categorization: A curriculum is an ordered set of courses; a course is an ordered set of batches; a batch is a set of trials. A trial is an instantiation of the input space, here, either a DD or an EA task. Parameterization of the input space as task parameters lets us conceptualize each course as a subset of the input space as well. For example, if we consider all possible trials of the DD task, one subset–or course–could consist of all trials with a 1s delay. Our goal is not to learn the entire input space, but rather a subset, which we call "desired task" When there is only one course (i.e., when the desired task is trained directly), we call this the *null curriculum*.

We extend the above definition of a curriculum to also include administration rules for different courses. These rules determine the number of batches in a course presented to the RNN during the curriculum as well as the transition between courses. For example, each course in a curriculum could be administered for 50 batches before transitioning to the next course in the ordered set. Here, we instead include a test set for each course that allows us to evaluate performance after each batch. Our test set is designed to have balanced discrepancy, with 10 examples of each allowable discrepancy. The final course of all curricula for a given task (EA and DD) share the same test set. If the RNN correctly solves 75% of tasks in the test set, it "graduates" to the next course. This performance threshold is chosen to allow failure on tasks with low discrepancy but many cues which are known to be especially challenging (Dehaene et al., 1998). An RNN output is correct if the integral over the decision period is within 50% of the integral of the target function.

All curricula are inspired by real world shaping procedures in experimental neuroscience, like those seen in Pinto et al. (2018); Duan et al. (2015); Stine et al. (2020); Romo et al. (1999); Constantinidis et al. (2018). We use the following three curriculum types:

**Evidence elongation curricula**: The first course has a cue period of 0.1s, which elongates by 0.1s per course until it reaches the length of the desired task's cue period. In EA tasks, this is

**Delay elongation curricula**: The first course has no delay period; delay elongates by 0.1s per course until it reaches the length of the desired task's delay. Only applies to DD tasks in which the final delay is 0.5s.

**Discrepancy reduction curricula**: The first course has a discrepancy threshold of 15 in EA tasks and 5 in DD tasks, which decreases by 1 with each course.

## 2.6 CURRICULUM COMPLETION TIME

We calculate curriculum completion time (CCT) by counting the number of batches (in our case, the same as the number of weight updates) a network sees during all the courses in a curriculum. Given the difficulty of our tasks, not all networks learn to solve the task in a reasonable time, and sometimes not at all, especially under a null curriculum (Fig 3). We set a weight update limit of 500 iterations. For comparing CCTs between "partially undefined" sets, we use rank-ordered Mann-Whitney U tests (Hettmansperger & McKean, 2010). RNNs that fail to learn can therefore still be included because their CCT must be greater than the CCT of all that do learn within the limit.

## 2.7 STATE SPACE ANALYSES

In neuroscience, dimensionality reduction techniques, e.g., Principal Component Analysis (PCA) are commonly used to infer dominant features of neural dynamics and to evaluate complexity (Cunningham & Byron, 2014). We similarly characterize and compare state spaces of our RNNs under different curricula and learning rules, and measure their effective dimensionality using PCA. We use all RNN activations from the final test set of each curriculum for computing the covariance matrix for PCA. The covariance matrix has trial time-steps × number of trials in the rows and neurons $N = 350$ in the columns. The resulting eigenvectors or PCs have $N$ components, and points along this PC space represent times during a trial as a "trajectory". As each time in a trial has a discrepancy (as per section 2.3), we can color PC trajectories accordingly (Fig 5AB).

We fit the first PC as the hyperbolic tangent of discrepancy. Formally, our eigendecomposition gives the $N$-dimensional state $S = \{s_i\}_i^N$ as a function of time $s_i(t) = p_{ij}\mathbf{a}_j(t)$, where $P_i$ is the $i$th PC vector with elements $p_{ij}$, $j$ iterates neurons, and $\mathbf{a}_j(t)$ is the activation of neuron $j$ at time $t$. We find parameters $(a, b)$ to maximize the correlation between $s_1$ and $\hat{s_1} = b + a\tanh(D(t))$. To report dimensionality, we measure the inverse participation ratio of PC eigenvalues (Rajan et al., 2010).

## 3 RESULTS

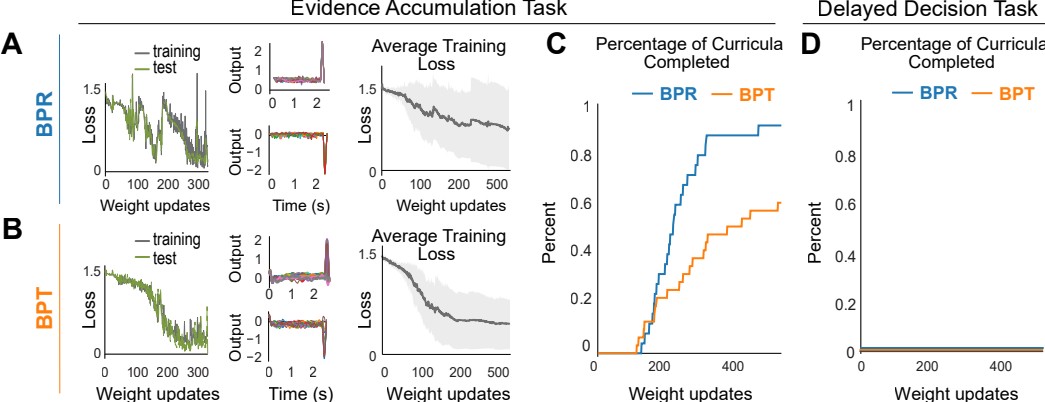

Figure 3: **Learning Without Curricula:** "Null curricula". **A, B:** Performance of BPR and BPT during training with the null curriculum on the EA task. (left) Green loss curves are the mean loss on the test set, whereas gray is the mean loss on the training batch. (middle) Sample network outputs on the test set. Left trials and right trials are separated by upper and lower panels respectively. (right) Average training loss over 50 RNNs trained. **C:** Cumulative density of curriculum completion times (CCT) of 50 RNNs with BPR (blue) and BPT (orange) in the null curriculum. **D:** Cumulative density of CCT with null curriculum on the DD task. All RNNs failed to pass the performance threshold.

## 3.1 NULL CURRICULA

Both learning rules are able to learn the desired EA task without curricula. BPR makes learning faster and more reliable, with a median CCT of 200 and all 50 RNNs successfully solving >75% of

the test set trials. BPT had a median CCT of 325 and only 32 of 50 RNNs learned within the 500 weight-updates limit. In Fig 3, we show that training and testing losses in the decision period largely overlap, with less noise in test loss. This is representative of the balanced discrepancy in our test set, whereas individual training batches are generated with random discrepancy. The decision period loss curve for BPR is steeper than that for BPT, demonstrating the benefit from representational loss.

For the desired DD task in null curricula, both learning rules fail to learn the task within the 500 weight update limit. Loss functions during the decision period, Fig 3, indicate little to no improvement in performance over training. Errors from the decision period cannot propagate through the long delay to the relevant directional cues.

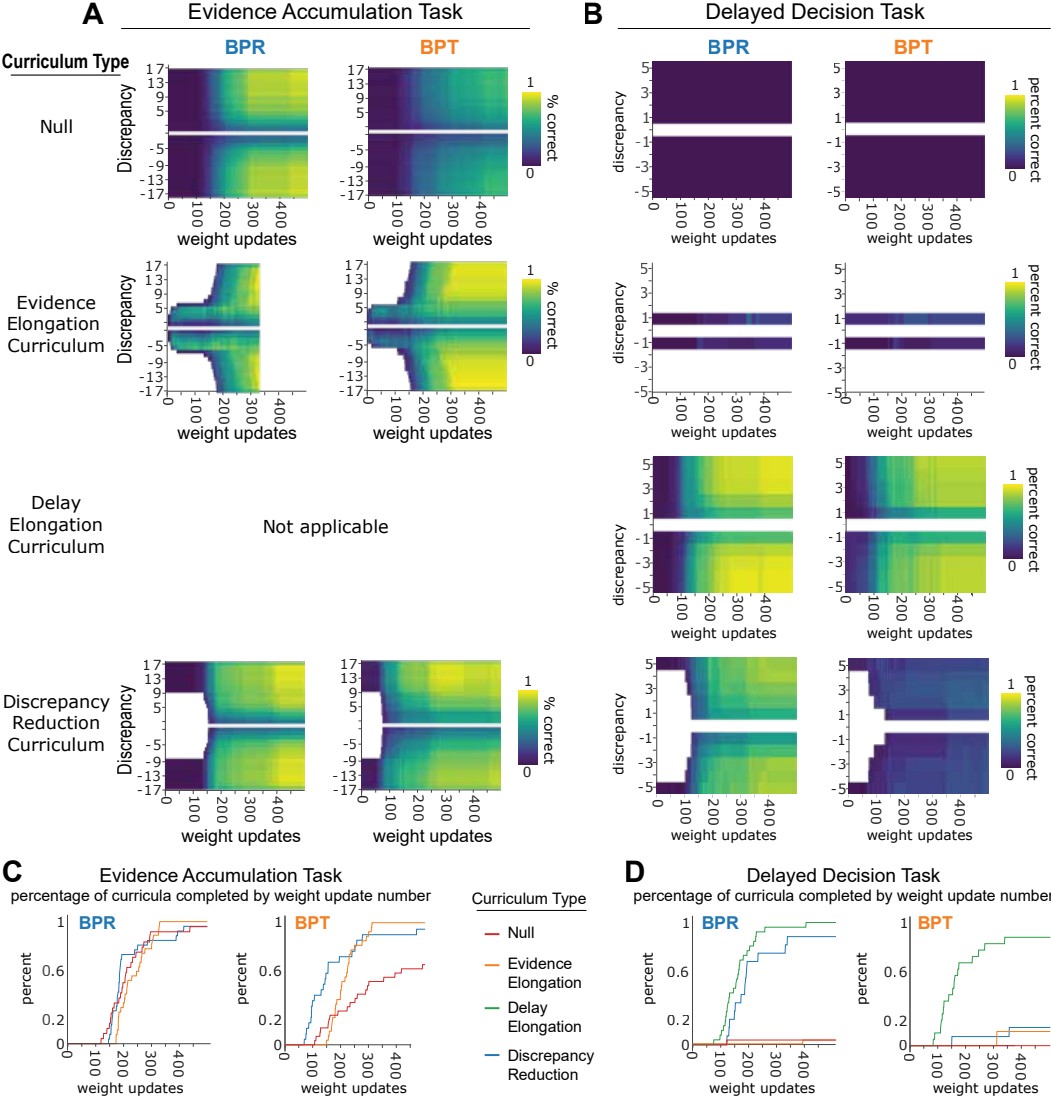

Figure 4: **Curriculum Learning Performance. A:** Evidence Accumulation (EA) task performance with BPR and BPT under all curricula. Each row shows a different curriculum and columns show BPT vs BPR learning rules. **B:** Delayed Decision (DD) task performance with BPR and BPT under all curricula. All heatmaps show average performance as a function of discrepancy and weight update. Each pixel corresponds to the average performance of 50 networks on a single discrepancy. Pixels are excluded if there are fewer than three RNNs tested on that discrepancy. As RNNs may be in different courses at any given time, if a discrepancy is not yet in a particular RNN's test set, that RNN's performance is assumed to be zero on that discrepancy. **C, D:** Cumulative density of CCT under all curricula. Each use 50 RNNs and show the % of RNNs which have completed a curriculum within a particular number of weight updates.

## 3.2 CURRICULUM LEARNING

Curriculum learning increases learning speed and reliability in the EA task for BPT (Fig 4). However, there is no obvious benefit from curricula for BPR in the same task (Fig 4). For the DD task, only the delay elongation curriculum successfully rescues BPT. In BPR, learning is significantly faster during both discrepancy reduction and delay elongation curricula (Fig 4). Evidence elongation curricula are unable to rescue the performance of RNNs trained by BPR, and fail to learn within 500 updates.

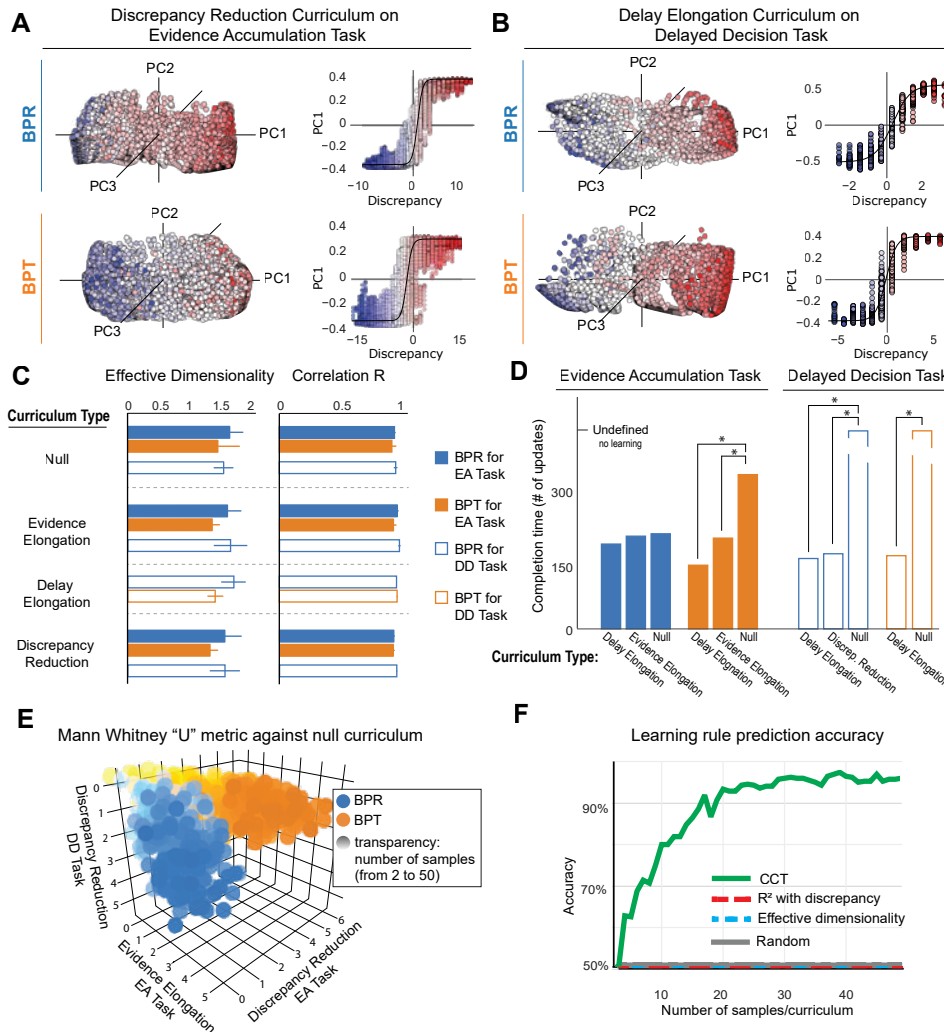

Figure 5: **Discriminating principles by learned state versus curriculum completion time. A:** (left) Low dimensional state spaces of networks trained by BPR and BPT with a discrepancy curriculum on EA task. Principal component (PC) scores of each time during the test set shown on the largest three PCs. Each point is colored by the discrepancy $D(t)$, with blue being the most negative discrepancy; red, the most positive. (right) Fit between $\tanh(D(t))$ and PC1. **B:** Low dimensional state spaces on delay elongation curriculum on DD task. **C:** Column 1: Effective dimensionality of BPR (blue) and BPT (orange) under each curriculum (row) for EA (filled bars) and DD tasks (open bars). Column 2: Correlation R between the first principal component and $\tanh(D(t))$ for BPR (blue) and BPT (orange) under each curriculum. **D:** Comparison between curriculum completion times (CCT) for each task (open vs filled) and curriculum type (x axis). Significance asterisk indicates $Z > 2$ or $p < .05$. **E:** 3D scatter plot of Mann Whitney "U" metrics. Each axis shows a different curriculum compared against the corresponding null curriculum. Each point is bootstrapped with the number of samples corresponding to the transparency (from 2 to 50 samples). Blue circles are from RNNs learning with BPR; orange, from BPT. **F:** Prediction accuracy with bootstrapped data, averaged over 1000 instantiations.

### 3.3 LEARNED STATE SPACES INVARIANT TO TASKS, CURRICULA, AND LEARNING RULES

Despite learning with different loss functions and under diverse curricula, state spaces of all successfully trained RNNs have the same key features. In particular, all are low dimensional, with two PCs explaining $\geq 90\%$ of the variance. Effective dimensionality is approximately 1.5 in all cases (Fig 5). Further, the first PC of all state spaces is strongly correlated with the hyperbolic tangent of the discrepancy, $\tanh(D(t))$. The correlation between the data and our fit is $\geq 0.9$R in all solutions. We argue that given the similarity of the spaces in the learned state, differentiating between learning principles is non-trivial and will represent a key advance for neuroscience.

### 3.4 RELATIVE CURRICULUM COMPLETION TIME IS INDICATIVE OF LEARNING PRINCIPLE

In the EA task, evidence elongation and discrepancy reduction curricula significantly change the distribution of curriculum completion time (CCT) only for BPT. Evidence elongation curricula significantly speeds up learning with a median CCT of 190 vs 325 weight updates (Z: 2.9 from U Test). Discrepancy reduction curricula make learning even faster with a median CCT of 134 (Z: 3.7). For BPR, learning is not significantly improved by any of the curricula tested.

For the DD task, we observe that delay elongation curricula benefit both BPT and BPR, but only BPR benefits from discrepancy reduction curricula. For discrepancy reduction curricula, BPR has a median CCT of 226, which significantly faster than under null curricula that fail to converge in our 500 weight-updates limit (Z: 3.9). Delay elongation curricula were even faster with a median CCT of 150 (Z: 5.4) with BPR. Similarly, for BPT, the median CCT was 154 with significance of Z: 4.16.

Our results in Fig 5 suggest that using the CCT distribution provides enough information to differentiate our representational learning (i.e., BPR) versus target-based (i.e., BPT) models. We evaluated this on bootstrapped data (Fig 5E, F). As the number of RNNs for each curriculum increases above 20, we see that CCT performance improvement from curricula successfully disambiguates BPT and BPR. Using a simple logistic regression on comparisons between CCT, we are able to identify the learning principle in over 90% of samples with 20 RNNs. Performing an analogous test with the standard neuroscience state space features, however, was unsuccessful even using the full dataset.

### 3.5 DISCUSSION

We proposed a novel use for curriculum learning as a method for uncovering the learning principles of a system. We demonstrated using model neuroscience tasks and shaping-inspired curricula that information about learning principles are inaccessible by studying only fully trained systems. However, we suggest that using only behavioral data we can uncover this information using curricula. This motivates the collection and curation of behavioral data (and eventually, concomitant neural data) during broad range of shaping procedures already being employed in labs.

Our approach focuses on an easily accessible metric of performance–the time to complete a curriculum. It is likely that other behavioral metrics such as a moving average of performance during curricula could also be valuable or provide better resolution in uncovering learning principles, particularly when combined with neural findings. While we demonstrated that global features of learned neural activations were largely invariant to our selected tasks, curricula, and learning rules (Fig 5, tracking the time evolution of activations (and weight matrices) during the execution of different curricula is likely to be a rich source of information about learning principles. In Nayebi et al. (2020), the authors successfully use information from activations during training without curricula. The addition of curricula to their approach could further improve learning rule discrimination.

While we demonstrated the value of our approach in task variants of evidence accumulation and delayed discrimination, we suggest that using curricula to discriminate learning principles could be more general. In other words, curricula could be designed to separate learning principles in the context of other tasks as well. Furthermore, our tasks are qualitatively similar to those used in human studies investigating numerosity (Testolin et al., 2020; Creatore et al., 2021). The noninvasive nature of our approach could be particularly advantageous where ethical or technical considerations limit our access to neural data, e.g., in humans. We hope that this paper will encourage other AI/ML and computational neuroscience researchers to expand our approach to more tasks and more curricula to further discriminate learning principles. We expect this work to result in more such effort to provide higher-resolution insights into learning principles in the biological brain and motivate better data collection from different behavioral experiments.

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

## ACKNOWLEDGEMENTS

This work is supported by NIH, NSF  McDonnell Fdn grants to KR.

