# OpenReview forum: "Curriculum learning as a tool to uncover learning principles in the brain "
_ICLR.cc/2022/Conference — ICLR 2022 Poster_

### Official Review · Reviewer_Nbow · 2021-10-25

**Correctness:** 3
**Technical Novelty And Significance:** 2
**Empirical Novelty And Significance:** 3
**Recommendation:** 6
**Confidence:** 4

**Main Review:**

- This research work is interesting and recent related work is adequately referenced.
- The introduction and discussion sections are well-written and clearly motivate the research work and its potential impact. Vice versa, the methodological and result sections are very concise and often hard to read. I suggest rebalancing the text, by expanding and clarifying the presentation of results.
- There are lot of hyper-parameters that seem to have been arbitrarily chosen (e.g., number of recurrent units, weights initialization schemes, events’ distribution, periods’ length, weight of representational loss, learning rates, accuracy percentage required to pass to next course, etc.). What was the rationale for setting all these parameters? How robust are the results to variations in these parameters?
- I understand the idea (and the importance) of adjudicating between different “learning principles”. However, I am not fully convinced that the two regimens implemented here (BPT vs. BPR) faithfully represent the difference between target-based learning and representation learning: in both cases, the feedback provided is tightly related to the feature that needs to be discriminated (i.e., number of events), which makes both regimens quite biased toward a target-based learning condition.
- Discrepancy is of course important for the task being simulated. However, I wonder whether the ratio between the number of cues should also be considered, since it should be easier to discriminate 5 vs. 6 compared to 15 vs. 16, though the discrepancy is the same -- see “size effect” and Weber’s law in psychophysics [1].
- The focus here is mostly on neuroscience experiments where animals sequentially collect evidence (number of cues) during task execution. However, the approach proposed by the authors might also fit well with other recent frameworks that have been used to simulate numerical discrimination, where we could differentiate between target-based learning (i.e., the goal is to classify input numerosity [2]) or representation-based learning (i.e., the goal is to first build a generative model of the environment, and then eventually learn discrimination tasks [3]). Discussing these similarities could help in building useful cross-disciplinary bridges.
- Readability of Fig. 3 could be improved. One solution would be to simply remove panels D, E and F, which contain “null” redundant information (learning is unsuccessful in both BPR and BPT) that can be easily embedded in the main text.
- Also readability of Fig. 4 could be improved, along with the related discussion in Sec. 3.2. My guess is that the authors mistakenly swapped the bottom summary panels (EA vs. DD)?

References
1. Dehaene S, Dehaene-Lambertz G, Cohen L. Abstract representations of numbers in the animal and human brain. Trends Neurosci. 1998;21: 355–361. doi:10.1016/S0166-2236(98)01263-6
2. Creatore C, Sabathiel S, Solstad T. Learning exact enumeration and approximate estimation in deep neural network models. Cognition. 2021;215: 104815. doi:10.1016/j.cognition.2021.104815
3. Testolin A, Dolfi S, Rochus M, Zorzi M. Visual sense of number vs. sense of magnitude in humans and machines. Sci Rep. 2020;10: 1–13. doi:10.1038/s41598-020-66838-5

**Summary Of The Paper:**

In this paper, the authors propose to use performance measures collected during curriculum learning as a way to adjudicate between different learning principles that might support (animal) behavior during a specific task. To this aim, the authors show that the learning principles governing two systems (RNNs) trained according to different objective functions can be inferred by examining the curriculum learning dynamics, despite the final “state” (effective dimensionality) of the networks is indistinguishable.

**Summary Of The Review:**

Though the focus of this work is mostly on computational neuroscience, the research scope might be broad enough to be of partial interest to the audience at ICLR. However, the research has some weak points, and the conclusions are supported by a single experimental setup, thereby undermining the overall strength of the paper.

------
After revision, I update my score from 5/10 to 6/10.

---

> ### Author Response · Authors · 2021-11-23
> **response to Nbow**
>
> Thank you for your time and effort reviewing our work. We appreciate your insightful comments, and have addressed your concerns here and in the updated manuscript.
>
> With regards to hyperparameters, we have more adequately explained in the text the following hyperparameter choices:
> 1) For our weight initialization scheme, we follow Sompolinsky et al, 1988. This enables use to avoid networks that are chaotic at initialization by using a scaling factor, g=1. We have updated the text to indicate this in Section 2.1
> 2)Our event distribution and period lengths are set to be analogous to mouse experiments from Brody et al 2018. We added this information to Section 2.2.
> 3) We weigh the representational loss to be on the same order of magnitude as the target loss.We have added this information to the text in Section 2.4 when we introduce this loss function.
> 4)Similarly, our learning rates are proportional to our weight initialization. This has also been added to Section 2.4 for clarity.
> 5) We have added a section to explain the accuracy threshold of our curricula in Section 2.5. We use 75% to allow for failure on tasks with low discrepancy but many cues. As you mention in another part of your review, these tasks are especially challenging and we allow networks to get them wrong.
>
> In general, our results are insensitive to most parameters. In particular, we have tested number of units much smaller (50, 100, 150, 200) and larger (500, 1000 units) and find consistent results. Initialization is highly flexible as we train with Adam optimization which is known to overcome poor initializations quickly. As for task parameters, most are selected by analogy to mouse experiments. In order for curricula which shorten period lengths to be beneficial, periods (such as evidence or delay) must be long enough for the task to be difficult. We selected out task period parameters conservatively to ensure we were well above the point in which the task becomes too easy to benefit from curricula.
>
> Thank you for pointing out the bias towards target-based representation. This is a limitation brought on by using a behavior matching task.  As this is central to our goal of neuroscience experiment, this is a necessary limitation that both models must include target learning principles. We have updated the text to reflect this, please see Section 2.4, in which we clarify that we are actually detecting the presence or absence of representational constraints. In other words, we cannot test the case where the model follows a purely representational-only-learning principle, but rather the presence of representational learning in addition to target learning.
>
> You are absolutely correct that the number of cues and ratio of signal is relevant to task difficulty. We have updated the task to include this, please see Section 2.5. Furthermore, your comment highlights that even in evidence accumulation and delayed decision tasks, there are many more possible curricula that could be designed (such as ratio-based or log-based curricula) to get further resolution into learning principles.
>
> We thank you for bringing the research on numerical discrimination to our attention. You are correct that this could help in building cross-disciplinary bridges, and we have added to the Discussion section and cited the related work.
>
> We have also significantly reworked Fig 3 and Fig 4 according to your recommendations. In Fig 3 we have removed panels E and F to reduce redundancy and improve clarity. In Fig 4 we have corrected the mislabeled panels and aligned rows to correspond to curriculum type.

---

> > ### Comment · Reviewer_Nbow · 2021-11-23
> > **Slight improvement of the paper, possibly making it of sufficient quality**
> >
> > After reading the authors' responses and the new version of the manuscript, I think that the quality of the paper has slightly improved. The authors addressed most of my criticisms, though some weaknesses are intrinsic in the proposed approach (e.g., bias towards target-based representation) and cannot be easily improved. The authors could also have analysed more carefully the case for a ratio effect, besides the discrepancy variable. In my view the revision makes the paper of sufficient quality (I raise my score to 6/10), though the overall impact and relevance for ICLR might still be somewhat limited.

---

> > > ### Author Response · Authors · 2021-11-29
> > > **additional response to reviewer Nbow**
> > >
> > > Thank you very much for your time and careful review.
> > > We believe that through your constructive criticism and insightful feedback, our paper has improved and we are grateful that you have updated your score to reflect that.

---

### Official Review · Reviewer_cUgC · 2021-11-01

**Correctness:** 3
**Technical Novelty And Significance:** 3
**Empirical Novelty And Significance:** 4
**Recommendation:** 6
**Confidence:** 3

**Main Review:**

### Strengths
1. The paper is written very clearly, text and figures are easy to follow.
2. The idea of using the learning time is new
3. Experiments are well designed with 2 different but commonly used neuroscience tasks, and 3 different curriculum types.
4.  The comparison of the learning process (loss function) on the basis of dimensionality, principal components, and state-space shows that it is difficult to disambiguate the learning process based on final representations thus serving as suitable controls to emphasize the comparison using curriculum learning time.


### Weakness
1. The plots in Figure 4 seem incorrect. In Figure 4A, in null curricula, the RNNs are unable to learn EA tasks while this is not the case in Figure 3 and as reported in the text. It seems that EA and DD columns are swapped.
2. There are a few typos such as in section 3.2 unnecessary [] are present in the text.
3. No caption for Figure 5E, what is the meaning of the asterisk (p < ?).
4. In Figures 5 C, D, and E it is not clear what is compared to what and which values are significant or non-significant. Is evidence accumulation compared with the delayed decision or BPT compared with BPR? I encourage the authors to think about how to make the plots clearer so that the plots emphasize what is emphasized in the text. In the current state, from the Figures themselves, it is not clear.
5. What is H in equation 4?


Ignoring a few minor weaknesses, I believe the paper demonstrates the effectiveness of using curriculum learning time as a criterion to identify different learning processes very clearly, thus emphasizing the point to collect behavioral and neural data during the learning process to identify the learning mechanism in animals.


**Summary Of The Paper:**

In this paper, the authors propose an approach using curricula to identify how a system has learned. Using two commonly used tasks in neuroscience: evidence accumulation and delayed decision recurrent neural networks (RNNs) are trained using two different loss functions (target based and representation based). They show that by simply comparing the learning time during different curricula , we can identify which loss function was used. On the other hand, the learned state-space trajectories of RNNs are indistinguishable thus unable to disambiguate which loss function was used for training.

The findings here put emphasis on the collection of behavioral and neural data while animals in neuroscience labs undergo curriculum-based learning on top of the data that is collected after the animal has learned.

**Summary Of The Review:**

The idea of using the curriculum learning time as a criterion to disambiguate learning processes is novel and the authors have done a good job in clearly demonstrating that using a wise choice of tasks and curricula. However, the results are not conveyed to the reader correctly and can mislead the readers if presented in this form. Therefore, I encourage the authors to put some effort into improving the plots and correcting the mistakes.

Due to the above reasons, I have given it a rating of 6 but I believe the ratings can be increased if the authors address the weaknesses.

---

> ### Author Response · Authors · 2021-11-23
> **response to cUgC**
>
> Thank you very much for your review and detailed comments.
> 1) We are grateful for you spotting that Fig 4 was mislabeled. We have corrected it and restructured Fig 4 to improve clarity.
> 2) We have removed brackets in Section 3.2 according to your suggestions and have tried to identify and eradicate typos from the (now updated) manuscript.
> 3) We significantly redesigned all figures to improve clarity in response to your review. Fig 5, in particular, has been remade with restructured panels and some new panels.
> 4) All the figure panels are captioned and annotated to indicate that significant \* means Z>2 i.e. p<0.05.
> 5) We have indicated in the text in Section 2.3 that H is the Heaviside step function.

---

### Official Review · Reviewer_cCVe · 2021-11-02

**Correctness:** 4
**Technical Novelty And Significance:** 3
**Empirical Novelty And Significance:** 3
**Recommendation:** 8
**Confidence:** 3

**Main Review:**

strengths:

- novel idea to use curricula to find signatures of different loss functions
- curricula inspired by those used in neuroscience experiments - directly applicable. Recommends how (relatively) easily collected behavioural data could be used to infer loss functions being optimized during task training.

weaknesses:

- mostly relevant to neuroscience/psychology
- missing final analysis to really demonstrate claim in detail by using the CCT to adjudicate between candidate cost functions. Right now the main result seems to be in subpanel E of figure 5, but this is only describes that differences exist. What about an accuracy on actually using this difference to predict cost functions in some simulation comparable to a neuroscience/psychology experiment? I'd recommend moving some of the other figures/sub panels to the supplemental to make room for such an additional analysis.
- The motivation for the two cost functions and how they represent different learning "principles" could have been clearer.
- The presented analysis only shows how the CCT distinguishes the particular two cost functions evaluated here. As far as I can tell, the authors don't propose that behaviour under different curricula can say something generic about the learning principles at play. It's not clear whether CCT would distinguish between two arbitrary cost functions or how it would help infer learning principles in the absence of clear hypotheses.

minor comments:

- Figure 4 is very useful. It makes the results much easier to grok than in the text. It would be better if two rows corresponded to curricula. Right now they are not matched.
- check wording in 2nd last paragraph. not sure what you mean to communicate here: "Furthermore, we suggest that relative predictions, such as the ones we make from curriculum completion times in Figure 5, are invariant to shifts in their absolute quantity might that arise from extending a model to a real brain"

**Summary Of The Paper:**

The paper shows how comparing network performance under different training curricula can distinguish learning principles. The curricula evaluated are inspired by those actually used in neuroscience to train animals to perform decision making tasks, and so the results motivate the analysis of behaviour during training periods which could provide evidence for/against candidate cost functions.

**Summary Of The Review:**

I recommend acceptance because the novel demonstration that behavioural differences under different curricula can be used as a signal to compare candidate loss functions is valuable. However, the demonstration seems somewhat narrow in scope and it's not clear how useful it would be in practice, so my recommendation is not a strong one.
**********************
EDIT: updated my score from 6 to 8 based on author response and changes to the paper

---

> ### Author Response · Authors · 2021-11-23
> **response to cCVe**
>
> Thank you for your time and insightful review.
>
> In response to your review we have added two panels to Fig 5 to demonstrate more definitively our claim of separating learning principles. In Fig 5, we have used bootstrapping and a simple logistic regression to show how many samples are necessary to make this deduction. Furthermore, we show that an analogous logistic regression using learned state space features is unsuccessful. Please see Fig 5 and Section 3.4 in the updated manuscript.
>
> We also have updated Section 2.4 to provide more clear motivation for the two cost functions. In particular, target-based loss (BPT) provides an idealized output behavior and representational loss (BPR) provides an idealized representation. In other words, BPR represents the case where the network has a representational constraint on its activations which facilitates task completion. To further increase confidence in this form of BPR, we also mention in Section 2.4 other forms of representational loss that produce the same CCT profile as BPR. The two other representational loss functions we consider are: a standard triplet loss and a loss function which encourages non-zero fixed points. These formulation lend to task completion only indirectly but produce the same CCT profile.
> In that sense, our task and curriculum design does not arbitrarily separate learning rules, but instead is for specific detection of the representational learning principle. Other tasks and curricula might be designed in the future to further disambiguate other principles or perhaps even arbitrary learning rules. This is an important highlight of our work: although our results have immediate value to neuroscience, our general approach has the promise to open a new kind of analysis for interrogating learning systems using curricula.
>
> Finally, we appreciate your comments on Fig 4. Accordingly, we have now updated the rows to correspond to curriculum type and increased figure clarity overall.

---

> > ### Comment · Reviewer_cCVe · 2021-11-28
> > **An interesting, novel idea whose impact remains to be seen**
> >
> > Thank you for your response. I've looked over the updated figures/sections you mentioned. The new panels added to Fig 5 greatly strengthen the presented claims. The definitions provided in 2.4 make it easier to interpret your use of terminology and the specific claims made. You suggest that the general framework may eventually be used to distinguish between other learning principles, but the presented work only provides a way of distinguishing specifically between target-based and representation-learning principles. The methodology does not distinguish specific learning rules or cost functions. Cost functions that fall under the same "principle" behave similarly in this analysis.
> >
> > I still think this work presents a sufficiently interesting idea to merit acceptance. But it remains to be seen what impact this work will have. Will a comparable experimental design be feasible in animals/humans? What if learning in biological brains is neither target-based or representation-based, but rather based on some other learning principle. What information will behaviour collected during curricula offer then? I tend to agree with reviewer wnW4 that being very explicit about what would be necessary for this to be applied to animal experiments would maximize the chance that it is taken up by neuroscientists.

---

> > > ### Author Response · Authors · 2021-11-29
> > > **additional response and clarification for reviewer cCVe**
> > >
> > > We appreciate your additional response.
> > > Thank you also for your kind words on the new Figure 5 and your review which resulted in strengthening our presented claims.
> > > We worked hard to address your specific concerns and we hope your score will reflect your improved opinion of the paper.
> > >
> > > You bring up an excellent point about feasibility in animals/humans and how to deal with unexpected results.
> > > As far as feasibility, shaping is already being performed in nearly all experimental neuroscience labs interesting in learning, decision making, and working memory mechanisms.
> > > In communications with several such labs, we asked if they could curate data on the shaping procedures they were already using, including from multiple shaping protocols they have tried (analogously to the different curricula we model), those that failed outright, or took too long.
> > > The responses were similar: "show us that it would be useful and we will record it."
> > > As you mentioned, tasks and shaping procedures might not line up exactly with what we have demonstrated here.
> > > As real shaping data across different tasks becomes available, our modeling framework allows us to adapt the design of our curricula and adjust our tasks appropriately.
> > > This will allow us to evaluate learning principles (not just the two investigated here, but also other candidate principles) in models with tighter conjunction with those data.
> > > However, we need community motivation for better collection of behavioral data during shaping.
> > > This paper is the motivation -- we demonstrate using models how shaping data can be even more informative than data from fully trained animals, which are much more commonly targeted in neuroscience experiments.
> > >
> > > In summary, our goal as stated in the introduction of our paper is to motivate experimental labs to collect and curate shaping/curriculum data from experiments they are already doing.
> > > We do this with an example which shows that such data can tell us more about learning principles than neural data from a trained system, which is currently highly-prized in experimental neuroscience.
> > > In doing so, we not only demonstrate our ability to separate two learning principles of value to neuroscience, but --perhaps more importantly-- that such a discrimination was made possible from data most neuroscientists discard.

---

### Official Review · Reviewer_wnW4 · 2021-11-02

**Correctness:** 2
**Technical Novelty And Significance:** 2
**Empirical Novelty And Significance:** 2
**Recommendation:** 5
**Confidence:** 4

**Main Review:**

Strengths:

The general idea of using learning behaviour under different training curricula to decipher something about the underlying learning process of a system is really interesting.
Simulating classical neuroscience tasks was a nice starting point, as it creates  opportunities to compare behaviour of these RNN models to real rodent and neural data, however this ended up being a bit of a  missed opportunity as there was no comparison to real animal data in this paper, as far as I could see.

Weaknesses:

There was a lot of confusion around terminology in the paper. Learning rules vs loss functions was a regular miscommunication. At many points in the text (including in the abstract) they say that their networks use two different learning rules (and so suggest the paper is able to diagnose different learning rules), when really the paper uses two different loss functions: one which uses the sum squared error loss on behaviour and the other which uses this same loss + a representational loss term. The learning rule is the same for both RNNs, which is batched gradient descent with an Adam optimizer.

The paper does not at any point, as far as I can see, actually use the RNN behaviour to predict the underlying loss function empirically, so claims that the paper shows they can disambiguate the loss functions (in the paper called ‘learning rules’), are not empirically substantiated. The paper could do this using the set of behavioural CCT data.

One of the two particular choices of loss function used in this paper require knowing the RNN’s (in their case, but animal’s in their suggested use case) internal representations, however one of the big motivators for this work was to gleam understanding about learning processes without neural data. To even reproduce these exact experiments in an animal you would need the neural data.

Generally it’s difficult to do well-controlled curriculum experiments and how to evaluate them is not always clear. CCT alone is a bit of a tricky metric to rely on. Looking at results from a single loss function, the different course completion times are to be expected as the courses themselves have different numbers of stages (just as completing an undergraduate degree should take longer and have different CCT times than learning about a subject by reading an article on wikipedia). How exactly do the authors plan on using this data to disambiguate learning processes in animals? One possible approach which I did not see (although I may have missed it) explained in the paper would be to simulate learning in network models of animal learning under different curricula and learning rules,  and then compare these sets of behaviour to different sets of animals' behaviour trained on each of the different curricula, to determine which set of network behaviours most closely matches the different sets of animals empirically. I think perhaps this is what the authors are proposing, as it seems sensible, but it should be made clearer. However, note that this requires several groups of animals (one group per curriculum) in a between-subject experimental design, which of course requires its own data collection pipeline and it is not the case that this data could just be recorded as a bi-product from ordinary animal studies, in which all animals should have as close as possible to the same training curriculum, in order to ensure they reach the final trained state for their main neuro task (which is not guaranteed under different curricula).

Another way to compare behaviour would be to subject different networks (animals) to different curricula, and then assess the effectiveness of each of these curricula on some later task. The learning curves can be expected to be different, but performance on the final task might be expected to be better under some curricula than others.

It’s really unclear (generally) how to design curricula that lead to qualitatively different learning trajectories under different hypotheses about animal or network learning. While I am sympathetic to the paper’s goals, in practice this really limits the experimental use case for this approach.

A related paper that analysing biological learning dynamics to understand different potential underlying learning rules is Cao et al 2020. They show that the learning behaviour trajectory as well as the progression and final neural represeenation depend on the learning rule in linear networks (but dont use curricula to tease these apart): https://proceedings.neurips.cc/paper/2020/file/6275d7071d005260ab9d0766d6df1145-Paper.pdf


**Summary Of The Paper:**

This paper simulates simple RNNs performing two classic decision-neuroscience experiments (a free choice evidence accumulation task and a delayed decision evidence accumulation task). The paper examines learning behaviour of these networks under three hand-crafted curricula, for each of two different RNN loss functions. The paper claims that one can diagnose the underlying learning rule (really the loss function) an RNN is using based upon the learning behaviour observed across the set of training curricula. They suggest a similar approach can be used to identify learning rules in animal neuroscience experiments.


**Summary Of The Review:**

Interesting basic idea, but incomplete analysis to substantiate the paper’s claims about how this might work in practice with animal data.

---

> ### Author Response · Authors · 2021-11-23
> **response to wnW4**
>
> Thank you for your response.
>
> First, we want to thank you giving us the opportunity to clarify our terminology about learning rules, principles, and updates. We understand that these words have a lot of different meanings depending on context and have now included our working definitions for clarity in section 2.4. In particular, we refer to learning principles as a higher order categorization of learning rules. We refer to learning rules as the function optimized by learning. The learning update is then the actual weight/synapse trajectory in accordance to the learning rule. In our case, we have two learning principles: target learning and representational learning. We have one exemplar learning rule for each principle, target-based backpropagation or BPT and representational backpropagation or BPR. Both have learning updates according to BPTT.
>
> Secondly, as per your suggestion, we have added panels E and F to Fig 5 to identify the learning principle of the network empirically, rather than our previous approach of stating that such an identification is possible.
>
> Thirdly, we want to clarify a misunderstanding about our approach that may have negatively impacted your scoring. The review mentioned that in order to reproduce our experiments in animals, we would need neural data. This is false, and in fact a key claim we make is that our approach uses only behavioral data to infer the learning principle. This is an essential strength of our approach, and as you yourself noted, a primary motivator. Our approach is to train RNNs using model learning principles, and then use the relative curriculum completion time improvements from designed curricula to deduce the learning principle. With analogous experimental data from animals, we could then argue for the presence of representational learning in the animal based only upon the behavior of the animal.

---

> > ### Comment · Reviewer_wnW4 · 2021-11-23
> > **Thanks for the clarifications**
> >
> > Thanks to the authors for their responses and additional work on the paper. I believe that identifying the learning principles of the network empirically is an important addition to the work and will definitely improve the paper. I have increased my score accordingly.
> >
> > Thanks also for clarifying the intended use case in the response. If the paper itself could make the exact proposed workflow clearer, in terms of exactly how it would work with a suggested set of animal experiments, that would make the contributions of the paper stand out a little more clearly. Perhaps a figure would be helpful? I still believe it would be necessary to run separate behavioural experiments with different curricula on different animals, not to just record behavioural data as it happens to be generated by an existing animal training pipeline in order to make this method work, so I think that point should be made in the paper. Unless of course the authors have a workflow in mind that doesnt require this and that I haven't thought of. Either way, some clarity on this front in the paper would make the value of this method stand out.

---

> > > ### Author Response · Authors · 2021-11-29
> > > **Further response and clarification to reviewer wnW4**
> > >
> > > Thank you for your additional comments and updated score.
> > >
> > > First, you are correct that to directly implement our example in mice one would need to use different mice for each curriculum.
> > > As you can see in Fig 5F, we demonstrate that one could expect to disambiguate learning principles with 20 mice with each of 3 curricula.
> > > This is not a trivial exercise; however, it is well within the means of modern neuroscience.
> > > For instance, Brody 2018 paper that inspired our task design uses 100s of mice in a delayed discrimination task.
> > > However, in the above paper there is no shaping data; no failed or slow curricula were reported, only the most successful shaping protocol was used.
> > > Their analyses currently focus on state spaces of expert/trained animals, rather than using curricula and shaping data to probe learning principles as we propose in our paper.
> > > This is where our approach can add value, not just from a deeper dive into the above dataset, but in behavioral datasets from other labs training subjects--ranging from 100s of mice and rats, to 10s of non-human primates and 100s of humans--to perform complex, time-varying tasks.
> > >
> > > Second, we want to clarify a possible miscommunication.
> > > You mention in your comment that experimentalists would need to do more than just record what they are doing now.
> > > As described in our previous paragraph, that is true if one lab wishes to reproduce exactly what we have done in a vacuum.
> > > However, what we are suggesting -- and calling for in this paper -- is a more communal effort to curate and share shaping data.
> > > Most experimental neuroscience labs are using shaping procedures to train animals ranging from mice and rats to non-human primates or human subjects to perform different tasks, such as a decision making or navigation in virtual environments.
> > > The shaping protocols in use currently are generated effectively by trial and error and only a single, "best working" curriculum is published, usually as part of a methods' supplement.
> > > Our paper shows that knowledge of what curriculum was too slow or failed entirely is also highly valuable.
> > > In fact, our results [see for example, Fig 5 in the paper] even suggest that in some cases such "failure mode" data are more valuable than the currently favored approach of analyzing neural data from fully trained or expert animals!
> > > Further, when you consider that many labs are working on qualitatively similar tasks and using different shaping procedures, making communal shaping data available would facilitate more computational approaches based on curriculum learning like ours.
> > >
> > > As discussed in our introduction, neural approaches to disambiguate these learning principles are not currently tractable experimentally.
> > > This is because even large-scale neural recording modalities, e.g., wide-field imaging or neuropixel recordings in rodents, are very severely under-sampled.
> > > The partial-sampling problem is even starker for neural data from primate brains.
> > > Concretely, we do not yet know which brain area/s to record from, for how long, and at what sampling rate.
> > > However, we think that using models trained via curriculum learning like ours, analyzing the evolution of state space trajectories during the performance of different curricula and across tasks can provide a road-map for more targeted data collection in the future.
> > > That said, when longitudinal, high-spatial-resolution neural data collection becomes possible, there is no reason to expect they would require fewer animals considering the statistical challenges high-dimensional neural data bring.
> > >
> > > To summarize, while it could be valuable for one lab to reproduce our experiment exactly, that is not what we are calling for.
> > > Rather, we wish to demonstrate the broader value of collecting, curating, and publishing curriculum/shaping data based on experiments already being performed in labs worldwide.

---

### Decision · Program_Chairs · 2022-01-20

**Decision:**

Accept (Poster)

**Comment:**

In recent years, artificially trained RNNs have been used for studying systems and behavioral neuroscience in terms of their learned representations, dynamics, computation, and the learning process itself. This paper contributes to further identify learning principles that may be revealed by curricula. The proposed approach for finding signatures of different loss functions is a novel and interesting idea which is very well fit for the neuro-oriented ICLR audience and has potential impact in other fields.